# Body Dissatisfaction, Restrictive, and Bulimic Behaviours among Young Women: A Polish–Japanese Comparison

**DOI:** 10.3390/nu12030666

**Published:** 2020-02-29

**Authors:** Bernadetta Izydorczyk, Ha Truong Thi Khanh, Sebastian Lizińczyk, Katarzyna Sitnik-Warchulska, Małgorzata Lipowska, Adrianna Gulbicka

**Affiliations:** 1Faculty of Management and Social Communication, Institute of Applied Psychology Jagiellonian University, 30-348 Krakow, Poland; bernadetta.izydorczyk@uj.edu.pl (B.I.); katarzyna.sitnik-warchulska@uj.edu.pl (K.S.-W.); 2Faculty of Psychology, University of Social Sciences and Humanities, Hanoi 336, Vietnam; ttkha@vnu.edu.vn; 3Faculty of Psychology, SWPS University of Social Sciences and Humanities, 40-326 Katowice, Poland; octans@wp.pl; 4Institute of Psychology, University of Gdansk, 80-309 Gdansk, Poland; 5HireRight, 40-007 Katowice, Poland; ada.gulbicka@gmail.com

**Keywords:** body image, body dissatisfaction, bulimic behaviour, restrictive behaviour, eating disorders, socio-cultural standards, mass media, western populations, cross-cultural

## Abstract

The growing number of women, who are characterized by restrictive and bulimic behaviours towards their own body is observed especially in countries influenced by Westernalization. However, there is a lack of cross-cultural studies in this area. The main aim of the present study was to examine the psychological and socio-cultural risk factors for eating disorders in Polish and Japanese women. A cross-sectional research study was conducted among 18- to 29-year old Polish (*n* = 89) and Japanese (*n* = 97) women. The variables were measured using the Sociocultural Attitudes Towards Appearance Scale SATAQ-3, and the Eating Disorders Inventory EDI-3. The descriptive and comparative statistics, Spearman’s rho, and the stepwise regression analysis were used. The global internalization of socio-cultural standards of body image proved to be a significant predictor of Body Dissatisfaction among Polish and Japanese women. The main analysis showed a significant relation between the Drive for Thinness and Interoceptive Deficits in the group of Japanese women, as well as a correlation between Drive for Thinness and Asceticism in the group of Polish women. The obtained results could improve the prevention aimed the dysfunctional eating behaviours. However, the cultural nuances need to be considered in understanding the risk factors for eating disorders.

## 1. Introduction

### 1.1. Epidemiology of Eating Disorders

A psychological diagnosis of restrictive and bulimic behaviours towards the body is an important factor which indicates increased risk of eating disorders such as anorexia, bulimia, and compulsive overeating. Analysis of the source literature confirms the continual increase of the number of people suffering from eating disorders worldwide [1,2]. Epidemiological data confirm the increasing importance of the problem in European countries [3,4], including Poland [5,6], the United States of America [7], Latin America [8], Arab countries [9,10], Africa [11], and Asia [12,13,14,15,16], with the risk of development of eating disorders being especially high in Japan [17]. The Global Burden of Disease (GBD) study from 2015 [18] confirmed the steady increase of eating disorders in women all around the world [19]. The epidemiology of eating disorders around the world indicates that various forms of anorexia and bulimia are more common in the population of girls and young women [20,21], although data on the occurrence of such problems in men [22] and the elderly seem to underestimate the problem [23,24].

The growing number of young women and, increasingly, men who are characterized by self-destructive behaviours towards their own body through excessive and restrictive weight loss and physical activity, induced vomiting, and use of laxatives indicates that there is also an increasing need for scientific research into this phenomenon in diverse populations. 

The prevalence of symptoms indicates that the increasing risk of eating disorders is characteristic not only of women raised in Western countries, but also for people living in countries which experience industrialization, modernization and Westernization [2,25]. The research for this article was conducted on populations of Japanese and Polish women, being citizens of countries which have undergone intense processes of Westernization. The Westernization of Poland took place after the communist period [26] and in Japan the process of Westernization and industrialization began under the strong pressure of the West during the 19th and 20th centuries after the period of political and economic isolation during the imperial monarchy [27]. Stigler et al. [28] suggest that the changes resulting from Westernization are similar in different countries. However, there are differences in the way of implementing these changes.

### 1.2. Risk Factors for Eating Disorders

For ages standards of the “ideal beauty” to which people compared themselves used to be specific for every culture. Under the pressure of globalisation and westernisation process the national beauty standards in Asian, Arabic, European countries, as well as in Latin America and South Africa are blurred and unified [29]. Despite the globalization of appearance ideal, researchers still confirm cross-cultural differences [30]. Women from Central Europe countries, like Poland, report high level of ideal body-shape stereotype internalization [31]. In Asian countries, like Japan or Vietnam, drive for thinness is very intensive [32] but preferences of body proportions seem to be relatively culture specific [33]. 

A literature review confirms the significance of the socio-cultural impact of standards of body image promoted by mass media on the development of restrictive and/or bulimic behaviours towards eating in many countries of Europe, Asia, and North and South America [34,35,36,37]. The negative influence of socio-cultural standards of body image is especially visible in the population of girls and young women [21,38]. Two simultaneously-acting socio-cultural factors may contribute to the intensification of these behaviours: Internalization and pressure due to socio-cultural standards of body image and appearance. 

Furthermore, research must test a model with many variables—biological, physical, psychological, and socio-cultural—to identify the multifaceted influence of restrictive and bulimic behaviours on eating and the body [39]. In addition to physical (e.g., BMI—Body Mass Index) and socio-cultural risk factors, the psychological factors supporting the development of unhealthy eating behaviours are also worth mentioning. People who suffer from eating disorders often present comorbid disorders such as anxiety and affective disorders, self-harm, or abuse of psychoactive substances [4,40]. Chapuis-de-Andrade et al. [36] studied a population of Brazilians with an average age of 28.9 ± 8.7 years (women constituted 69% of the research group): The results showed significant correlations between compensatory (bulimic) eating behaviours and affective factors. Moreover, results obtained by Agüera et al. [41]—conducted on a Spanish population of 656 women and 62 men—also showed a significant relation between emotional dysregulation, personality, and psychopathology of eating disorders. 

Analysis of the literature suggests that the basic risk factors of eating disorders are directly related to the development of personal dissatisfaction with one’s body, excessive pursuit of thinness, and restrictive and bulimic eating behaviours, and this relationship is noted by authors conducting research in both Western cultures [1], including Poland [6,42], and Eastern cultures [13] such as Japan [17,43]. According to the literature from the last several years, people who suffer from anorexia or bulimia are dissatisfied with their bodies and are characterized by excessive perfectionism and a desire for control. They also may have interoceptive deficits, difficulties in building emotional bonds with other people, and suffer from other emotional disorders such as anxiety (including fear of puberty), depression, or obsessive-compulsive disorder [44,45,46,47,48,49,50,51,52]. Moreover, the following psychological factors are also associated with increased risk of eating disorders: Emotional dysregulation and low self-esteem [46]. 

### 1.3. Eating Disorders and Culture

The large number of results about the various risk factors for eating disorders indicates the need for further research on populations of various nationalities to identify which factors are universal and common to everyone and which are specific to particular cultural and social conditions.

Culture shapes the context in which attitudes towards the body are formed and, hence, it is a critical component to consider when trying to understand the development of body dissatisfaction and eating disorders [30,53]. 

There are many classifications and typologies of cultures [54], but attitudes towards the body differ most starkly between cultures divided into those of guilt and those of shame [55,56,57] or individualistic and collectivistic ones [58,59]. According to individualistic—collectivistic culture dichotomy, a person’s cultural identity may have an individualistic or collective character. The individualistic type focuses on their own needs, self, the importance of individual characteristics, and the independent existence of the person. However, the importance of the views, needs, and goals of the group is more important for the collectivist type. These behaviours focus on norms and obligations imposed by the collective community and emphasize the significance of attachment and commitment to cooperation [60].

### 1.4. The Current Study: A Polish–Japanese Comparison

Groups of young Japanese and Polish women were examined to conduct intercultural comparisons and to identify psychological and socio-cultural predictors of body dissatisfaction, restrictive behaviours (e.g., excessive pursuit of thinness), and bulimic behaviours. The cultural identity of Poles is individualistic. In contrast, the Japanese cultural identity is collectivist [61]. 

The aforementioned types of cultural identity also differ in terms of attitudes towards food among the Polish and Japanese. Polish people represent a “potato culture”, i.e., potatoes are dominant in their diets, while the Japanese represent a “rice culture”, where rice has a dominant place in their diet [61]. Paleczny [61] says “the selection of basic ingredients used in everyday diets and the means of preparation remains conditioned by cultural tradition”. In terms of attitude towards nutrition, the Japanese “rice culture” and the Polish “potato culture” are certainly different. This cultural difference is the basis of intercultural research to identify universal and cross-cultural predictors of eating disorders. In addition, the research cited in the introduction of this article indicates the potential presence of individual and intercultural differences which characterize various living conditions and the specifics of the upbringing of girls and young women in different cultures (i.e., “rice” and “potato” cultures). On one hand, despite potentially different cultural factors influencing the upbringing of young Polish and Japanese women, there are some cultural similarities due to the increasing internalization of the socio-cultural standards of Western culture promoted by the mass media in both countries [57]. Moreover, the process of Westernization is connected not only with “thin ideal of beauty” [62,63,64,65] but also by unifying dietary patterns in countries from different climatic and cultural regions [66], such as in Poland and Japan [62,63,67,68].

Therefore, the question is whether and to what extent the influence of the different traditions and cultures on the upbringing of women in Japan and in Poland may determine the significant similarities and differences in physical (BMI), psychological, and socio-cultural risk factors for developing body dissatisfaction and restrictive and bulimic behaviour towards the body and nutrition. 

### 1.5. Research Objective, Variables, and Research Questions

The research for this article was conducted on populations of Japanese and Polish women. The main purpose of this study was to identify which psychological and socio-cultural risk factors for eating disorders (i.e., dissatisfaction with the body and restrictive and bulimic behaviours towards nutrition and the body) are universal and which are specific to the cultures investigated. 

The first aim of this research was to search for the cross-cultural similarities and differences between Polish and Japanese young women in the strength of the relationship between body dissatisfaction, the level of restrictive and bulimic behaviour towards nutrition and the body, BMI values, a profile of selected psychological traits (suggested by the literature as typical for people with eating disorders), and levels of internalization and pressures of socio-cultural standards of body image promoted by mass media. 

The second aim of this research was to identify which predictors of dissatisfaction with the body and restrictive and bulimic behaviours towards the body and nutrition are universal and which are specific to Polish and Japanese cultures. 

Yoon et al. [69], emphasize in their research the importance of BMI in the development of disturbed eating behaviour in the group of modern adolescents. There are no comparative studies on predictors of body dissatisfaction and antihealth restrictive and bulimic eating behaviours in groups of Polish and Japanese women, hence we do not refer to other authors. Secondly, literature sources indicate that the socio-cultural and psychological variables presented in this article as potential predictors of restrictive and bulimic behaviours have been analyzed in other studies by various methods and most often only concerned the measurement of psychological traits profiles [46] or only focused on socio-cultural variables [70].

Risk factors for eating disorders were the dependent variables for this study, specifically: Body dissatisfaction and restrictive and bulimic behaviour towards nutrition and the body. These constitute a theoretical construct which describes potential indicators (based on the analysis of the literature and clinical criteria) of the development of anorexia or bulimia nervosa. These are: Body dissatisfaction, excessive restrictive behaviours (measured by strenuous pursuit of thinness), and bulimic behaviours towards eating [46]. The independent variable is psychological and socio-cultural factors which, according to the literature sources, can potentially explain body dissatisfaction and the development of unhealthy, overly restrictive, or bulimic eating behaviours. This independent variable is a complex factor, which consists of psychological factors, i.e., a basic psychological trait profile, relevant to the psychological diagnosis of people with eating disorders. These psychological factors are: Low Self-Esteem—an indicator describing the level of a person’s self-esteem.Personal Alienation—an indicator describing one’s level of reflectiveness and feeling of emotional emptiness.Interpersonal Insecurity Scale—an indicator describing the level of difficulty a person has expressing personal thoughts and feelings in the presence of other people and the strength of the tendency to isolate from others.Interpersonal Alienation—an indicator describing one’s level of disappointment, alienation, and lack of trust in relationships. It is also related to feeling trapped in a relationship. The person also has an inability to experience understanding and love from other people.Emotional Dysregulation—an indicator describing the level of intensity of mood instability, impulsiveness, recklessness, anger, and a tendency towards self-destruction.Interoceptive Deficits—an indicator describing one’s level of confusion in the accurate recognition of emotional states and stimuli which come from one’s body.Perfectionism—an indicator describing the intensity of the need for maximal accomplishment and the tendency to possess the highest possible standards for personal achievement.Asceticism—an indicator describing a person’s tendency to seek purity and virtue by striving for spiritual ideals such as self-discipline, self-denial, self-control, and self-restraint. It includes the control of one’s needs and drives. This factor also assesses positive connotations associated with achieving purity through restraint, guilt, and shame regarding pleasure.Maturity Fears—an indicator describing the level of a person’s desire to return to the safety of childhood. It is also related to the fear of psychosexual puberty.

Furthermore, the independent variable also comprises socio-cultural factors such as socio-cultural attitude towards the body, which reflects the level of internalization and pressures of socio-cultural standards of body image promoted by mass media. 

The second explanatory variable—socio-cultural factors—was defined as a theoretical construct describing the level of internalization and pressures of socio-cultural standards of body and physical appearance promoted by mass media. Moreover, it also describes the frequency of seeking information from mass media about physical appearance. An additional independent variable is BMI (Body Mass Index), measured by the following formula: Weight in kilograms divided into height in square meters. 

The following research questions were asked:(1)Is there a significant difference between the examined Polish and Japanese women in terms of the psychological traits investigated in the research model that are relevant for the development of the psychological profile of a person with anorexia and/or bulimia?(2)Is there a significant difference between the examined Polish and Japanese women in terms of their level of internalization of and pressure due to socio-cultural standards of body image promoted by mass media? Is there a significant difference between these groups in the frequency of seeking information regarding physical appearance in the mass media?(3)Does there exist and what are the main predictors of dissatisfaction with the body and restrictive and bulimic behaviours in the groups of Polish and Japanese women?(4)Is there a significant difference between Polish and Japanese women in terms of psychological and socio-cultural predictors of body dissatisfaction and restrictive and bulimic behaviours towards the body and nutrition?

## 2. Materials and Methods

### 2.1. Participants

The groups were selected through purposive sampling. The following inclusion criteria were applied: Age (18–30 years old), Polish or Japanese nationality, never having been treated for any mental disorder, lack of visible disability or distortion in physical appearance, and having lived with one or more parents, with or without siblings, from childhood to young adulthood (i.e., the beginning of the studies) in Poland or Japan for the respective groups. Criteria were verified by survey questions, which enabled the identification of exclusion factors. 

The research was conducted simultaneously in seven academic cities in Poland and in five, various, academic cities in Japan from 2018 to 2019. Qualified researchers conducted and coordinated the research simultaneously in Japan and in Poland. The study was conducted by email. The research groups were recruited from among the volunteers. The information about the possibility of participating in the study was propagated in students. Woman who met the inclusion criteria for the project were asked to invite acquaintances to participate; i.e., a nonrandom method of sample selection (“snowball sampling technique”)—the subjects were invited by email. Participants were women who were working or studying and who permanently live in Japan or in Poland. The purpose of the study was explained to all the research subjects. Individuals were asked for their consent to participate in the research and were informed that participation was voluntary and anonymous. 

The research plan was to examine 100 Polish and 100 Japanese women aged 18–30. However, a total of 186 women aged 18–26 participated in the final research: 89 Polish women and 97 Japanese women. Due to errors in competing questionnaires (incompleteness of obtained research data) and the participation of subjects of a nationality other than Polish and Japanese, the study included 70 Japanese women and 89 Polish women. The average BMI in the group of Japanese women was 20.20 and for the Polish group it was 21.79. The average age of the women examined was 22.36 for the Polish group and 20.66 for the Japanese group. The surveyed women in both groups were students of similar fields of study—humanities, social sciences, and medical sciences, as well as women who had already completed their education and were working professionally. The Polish women were living in cities which are academic centers: Warsaw, Kracow, Wrocław, Katowice, Bydgoszcz, Gdansk, and Poznań. The Japanese group also lived in the academic cities such as Tokyo, Kyoto, Yokohama, Nagasaki, and Nagano. Eighty two percent of participants were students, the rest (18%) were employed at the time of the study and had college education.

### 2.2. Ethical Approval

Ethical approval was obtained from the relevant institutional ethical review committees and the research was conducted in accordance with national and international regulations and guidelines. Written consent was obtained from all participants. The protocol of this study was approved by the Ethics Board for Research Projects at the Institute of Applied Psychology, Jagiellonian University in Krakow.

### 2.3. Methods

#### 2.3.1. The Eating Disorders Inventory—EDI-3 

The Eating Disorders Inventory—EDI-3 by Garner [46], in Polish adaptation by Żechowski [71] and Japanese adaptation by Shimura et al. [72], was used. The EDI-3 consists of 91 items organized into 12 primary scales: Drive for Thinness, Bulimia, Body Dissatisfaction, Low Self-Esteem, Personal Alienation, Interpersonal Insecurity, Interpersonal Alienation, Interoceptive Deficits, Emotional Dysregulation, Perfectionism, Asceticism, and Maturity Fears. The following three scales from EDI-3 were used to measure the dependent variable: Body Dissatisfaction (BD), Drive for Thinness (DS), and Bulimia (B). All scales have high rates of statistical validity and reliability in Polish and Japanese studies. Cronbach’s alpha values for the EDI-3 scales were as follows: Drive for Thinness (DS) = 0.86 (Polish studies), 0.90 (Japanese studies), Bulimia = 0.81 (Polish studies), 0.86 (Japanese studies), and Body Dissatisfaction = 0.92 (Polish studies), 0.86 (Japanese studies).

#### 2.3.2. The Sociocultural Attitudes Towards Appearance Questionnaire—SATAQ 3 

We used the Sociocultural Attitudes Towards Appearance Questionnaire—SATAQ 3 by Thompson et al. [70], in Polish adaptation by Izydorczyk and Lizińczyk [73], and a Japanese version of this test based on the adaptation by Yamamiya and Shimai [74]. The original version of the SATAQ 3 questionnaire consists of the four following scales: *Internalization-General* (describing the level of internalization of socio-cultural standards of body image; consists of nine items), *Internalization-Athlete* (a 5-item scale for measuring the level of internalization of athletic body shape), *Pressures* (describing the level of pressure of socio-cultural standards felt by a person; contains seven items), and *Information* (a 9-item scale which describes the frequency of seeking information about body image and socio-cultural standards of physical appearance). The SATAQ 3 questionnaire was completed by each subject by marking their answers on a 5-point Likert-like scale. The Cronbach’s alpha coefficients for the scales were as follows: *Internalization-General* = 0.93, (Polish version = 0.91), *Internalization-Athlete* = 0.80 (Polish version = 0.96), *Pressures* = 0.92 (Polish version = 0.78), and *Information* = 0.96 (Polish version = 0.89).

#### 2.3.3. Body Mass Index (BMI) 

To measure Body Mass Index (BMI), participants completed the survey with the following clinical data: Age, sex, body mass, and height. BMI was obtained by dividing the body weight in kilograms by the square of the height in meters. 

### 2.4. Statistical Methods

Statistical analyses were performed in Statistica 13.3 and in Excel (Microsoft Office 365 ProPlus). 

Stages of statistical analysis: 

Stage 1—descriptive statistics. Measuring the mean values of all variables in the research model. 

Stage 2—measuring the significance of differences between the average intensity of variables present in the research model in the group of Polish and Japanese women. In this stage of statistical analysis, the Mann–Whitney U test was used. 

Stage 3—measurement of the strength of the relationship between variables in the groups of Polish and Japanese women, which was followed by testing the significance of differences in the strength of correlation between the variables in both groups. In this stage, Spearman’s rank correlation coefficient (Spearman’s rho) was used. 

Stage 4—measurement of the strength of the relationship between the dependent and independent variables using stepwise regression analysis. The aim of this stage was to search for predictors of the dependent variables in the groups of Polish and Japanese women. 

## 3. Results

The selection of respondents was deliberate, so the groups did not differ in terms of age or BMI. 

### 3.1. Differences between the Groups

The next step of the statistical analysis of the collected data was to measure the significance of differences between the examined groups of Japanese and Polish women in terms of all variables included in the research model. Due to the fact that the studied variables were not normally distributed, nonparametric statistical tests were used in further analyses, specifically: Spearman’s rank correlation (Spearman’s rho) and the Mann–Whitney U test. The results of the analyses are presented below in Table 1. 

A comparative analysis of mean values of all variables in the Polish and Japanese groups showed the presence of significant differences between these groups in terms of certain variables: (1)Japanese women showed a significantly higher level of bulimic tendencies than Polish women. These results are significantly related to increased risk of developing eating disorders. This component is a risk factor for eating disorders (one of the dependent variables). Groups were not differentiated by other factors of this variable, such as Drive for Thinness or Body Dissatisfaction, which means that Polish and Japanese women show similar average intensity of these characteristics. However, the higher intensity of bulimic tendencies among Japanese women indicates that they more frequently experience thoughts about binge-eating, behaviours associated with emotional suffering, and inducing vomit to lose weight.(2)Japanese women showed significantly higher levels of Emotional Dysregulation than Polish women. This result indicates that they are characterized by higher levels of mood instability, impulsiveness, recklessness, anger, and self-destructive and impulsive tendencies. This can also mean the existence of potential problems with the abuse of psychoactive substances such as alcohol or drugs. Comparing this result with the higher intensity of the variable discussed in Section 1, it can be assumed that the group of Japanese women show higher impulsive and self-destructive tendencies than the group of Polish women.(3)Japanese women showed a significantly higher level of Asceticism than Polish women. It can be assumed that they more often show the tendency to seek purity through pursuit of spiritual ideals and exercise restraint due to guilt or shame (e.g., self-discipline, self-denial, self-restraint, self-sacrifice, and control of urges and bodily needs).(4)Japanese women also showed significantly higher levels of Maturity Fears than the examined Polish women. This indicates that Japanese women have a greater desire to return to the safety of childhood, which seems to be typical for adolescents whose weight loss is motivated by anxiety about psychosexual puberty. However, it should be noted that the average age of the Japanese women was 20.66 which means they are at the period of late adolescence. Polish women were a bit older, with an average age of 22.38.(5)In terms of socio-cultural variables, the only variable which significantly differed between the groups of Polish and Japanese women was frequency of seeking information about body image and physical appearance from mass media, with Japanese women doing so significantly more often. However, the level of internalization and pressures due to socio-cultural standards of body image promoted by mass media was similar in both examined groups.

### 3.2. The Relation between Studied Variables among Japanese and Polish Women

The next stage of the analysis of the collected data was to identify the presence of significant relations between the variables in both groups of Japanese and Polish women. This was done using Spearman’s rho coefficient. The obtained results are presented in Table 2. Analysis of the correlations between variables in the group of Polish women show that all significant dependencies were positive. By analyzing in detail the Polish part of the table below, we can observe: The presence of significant positive correlations of Information with Drive for Thinness and Asceticism. Significant and positive correlations of Pressures with Body Dissatisfaction, Perfectionism, Drive for Thinness, and Asceticism. Significant positive correlation of Internalization-Athlete with Body Dissatisfaction, Low Self-Esteem, Personal Alienation, Interoceptive Deficits, Perfectionism, Drive for Thinness, and Asceticism. Significant positive correlation of Internalization-General and all examined variables, except Interpersonal Insecurity.

To sum up, the strongest positive correlations were obtained between the global internalization of socio-cultural standards of body image and all variables except Interpersonal Insecurity, which describes the level of social isolation. It is also related to discomfort, anxiety, and social anxiety of the examined Polish women. In turn, the higher the internalization of socio-cultural standards regarding athletic body shape, the greater the Drive for Thinness, Perfectionism, Asceticism, Low Self-Esteem, and Self-Alienation. Less significant correlations were found between the pressures of socio-cultural standards on body image and the frequency of searching for information about body image and the Drive for Thinness, Body Dissatisfaction, Perfectionism, and Asceticism. The weakest significant and positive correlations were found between seeking information about body image and Drive for Thinness and Asceticism. 

Almost all correlations were also significantly positive in the group of Japanese women. The only significantly negative relationship was between Personal Alienation and seeking information about body image and physical appearance. The following details can be observed by analyzing the Japanese part of the table above: There was a significant positive correlation between Information and Body Dissatisfaction. Furthermore, there was also a significantly negative correlation between Information and Personal Alienation. There were significant positive correlations of Pressures with Bulimia, Body Dissatisfaction, Interpersonal Alienation, Interoceptive Deficits, Drive for Thinness, and Asceticism. There were significant positive correlations of Internalization-Athlete with Bulimia, Body Dissatisfaction, Low Self-Esteem, Perfectionism, Drive for Thinness, and Asceticism. There were significant positive correlations of Internalization-General with Body Dissatisfaction, Perfectionism, and Drive for Thinness.

To sum up, the most significant correlations were found between the pressure of socio-cultural standards on body image and all three components of risk of eating disorders, i.e., Bulimia, Drive for Thinness, and Body Dissatisfaction. Furthermore, correlation also occurred between socio-cultural pressures and Asceticism, Interpersonal Alienation, and Interoceptive Deficits. Moreover, the higher the Internalization-Athlete in the group of Japanese women, the greater the severity of all three risk factors for the development of eating disorders and the greater the intensity of such personality traits as Perfectionism, Asceticism, Low Self-esteem, and Interpersonal Alienation. The analysis shows the strongest positive correlation is between Internalization-Global and Drive for Thinness. Moreover, a weaker but also significant correlation was present between Internalization-Global and Perfectionism. Due to the fact that the correlation analysis revealed some differences in the strengths of relations between variables in the groups of Polish and Japanese women, we decided to conduct an analysis of differences between the correlations to determine their significance. 

The comparative analysis of correlation between variables in the groups of Japanese and Polish women showed the presence of four significant differences; they were highest for the correlations of Internalization-Global with Asceticism and Interoceptive Deficits (i.e., accurate recognition of internal emotional states and adequate response to them). In both cases, the correlation coefficient was higher for the group of Polish women than for Japanese women. This correlation was positive, which means that the stronger the trait of Asceticism and the greater the Interoceptive Deficits, the greater the level of global internalization of socio-cultural standards of body image and physical appearance promoted by mass media in the group of young Polish women. Asceticism is a trait that is also positively correlated with the socio-cultural Information variable. The correlations between these variables was also higher for the group of Polish women. This means that the greater the intensity of Asceticism (understood as the pursuit of spiritual ideals through self-discipline, self-denial, and self-control/restraint), the more often they seek information about body image and physical appearance. In turn, the higher the level of searching for information about body image, the lower the level of Personal Alienation (i.e., a sense of emotional emptiness, loneliness, and the lack of understanding of themselves and their feelings and thoughts) in the group of Polish women. Polish and Japanese women showed no significant differences in terms of other compared correlation coefficients.

### 3.3. Predictors of Dissatisfaction with the Body—Restrictive and Bulimic Behaviours Towards the Body and Nutrition among Polish and Japanese Women

In order to answer the third and fourth research questions, stepwise regression analysis was performed. The results of regression analysis for the groups of Polish and Japanese women are presented below in Table 3. The model verified on the basis of stepwise regression included only variables whose level of significance was < 0.05. Stepwise regression analysis was focused on the search for psychological and socio-cultural variables that could explain the risk factors for the development of eating disorders (Body Dissatisfaction, Drive for Thinness, and Bulimia). The values of adjusted R-squared and proportion of the explained variance in particular regression models were taken into account in the interpretation of the collected data.

The comparison of predictors in the studied groups showed a significant relation between the Drive for Thinness and Interoceptive Deficits in the group of Japanese women, as well as a correlation between Drive for Thinness and Asceticism in the group of Polish women (Table 3). 

### 3.4. Detailed Discussion

#### 3.4.1. Bulimia 

Based on the obtained adjusted R^2^ coefficients it can be concluded that for the Bulimia variable, Personal Alienation, BMI, Internalization-General, and Internalization-Athlete explained 31% of the variance of this dependent variable in the group of Polish women. The positive values of Beta coefficients confirm that the higher the level of Personal Alienation and the greater the intensity of global Internalization of the socio-cultural standards of body image promoted by mass media, the greater the tendency towards bulimic behaviour and bulimic thoughts in the group of examined Polish women. Furthermore, the values of the Beta coefficients also indicate that the higher the BMI, the greater the tendency towards bulimic behaviour and thoughts in this group. Other psychological and socio-cultural variables included in the research model turned out to be irrelevant in explaining bulimic tendencies as one of the risk factors for the development of eating disorders in the group of Polish women. 

In the Japanese group, it seems that 60% of the variance of the Bulimia variable was explained by: Interoceptive Deficits, Asceticism, Emotional Dysregulation, Interpersonal Insecurity, and BMI. The values of Beta coefficients of significant predictors confirm that the higher the levels of Interoceptive Deficits, Asceticism, Emotional Dysregulation, and BMI, the greater are the tendencies to present bulimic behaviours and thoughts in the group of examined Japanese women. Furthermore, values of Beta coefficients also indicate that the higher the Interpersonal Insecurity, the lower the tendencies towards bulimic behaviours and thoughts.

#### 3.4.2. Body Dissatisfaction 

Values of adjusted R^2^ coefficients for the Body Dissatisfaction variable indicate that five variables explained 59% of its variance in the group of Polish women. The values of Beta coefficients of significant predictors show that the higher the BMI, Interpersonal Alienation, Interoceptive Deficits, and Internalization-General, the greater the intensity of Body Dissatisfaction. Moreover, the more a person searches for information about body image and physical appearance, the lower the level of Body Dissatisfaction. Other predictors included in the research model—psychological and socio-cultural factors—proved to be not significant in explaining Body Dissatisfaction as one of the risk factors of the development of eating disorders in the group of Polish women. 

The obtained values of adjusted R^2^ coefficients for the Body Dissatisfaction variable indicate that four variables explained approx. 45% of the variance of the results in the group of Japanese women. Values of Beta coefficients show that the greater the Emotional Dysregulation, Pressures, and BMI, the higher is the intensity of Body Dissatisfaction in the group of examined Japanese women. On the other hand, the larger the Interoceptive Deficits, the lower is the level of Body Dissatisfaction. 

#### 3.4.3. Drive for Thinness

Based on the collected values of adjusted R^2^, it can be concluded that only two variables—Asceticism and Internalization-General—significantly explain approx. 62% of the variance of the Drive for Thinness variable in the group of Polish women. The calculations conducted using Fisher’s exact test and significant coefficients of factors estimated on the basis thereof show that the greater the Asceticism and the level of global internalization, the higher the intensity of the Drive for Thinness in the group of Polish women. Other psychological and socio-cultural variables included in the research model proved to be insignificant for explaining the variability of Drive for Thinness as one of the risk factors for the development of eating disorders in the group of Polish women. 

Based on the obtained values of adjusted R^2^ coefficients, it can be concluded that only three variables—Interoceptive Deficits, Internalization-General, and BMI—significantly explain approximately 54% of the variance for the results of Drive for Thinness in the Japanese group. Results of Fisher’s exact test showed that the higher the level of Interoceptive Deficits and BMI, the higher the level of Drive for Thinness in the group of examined Japanese women. In addition, the higher the intensity of global internalization, the lower the Drive for Thinness in the group of Japanese women. Other psychological and socio-cultural variables included in the research model proved to be irrelevant in explaining the variability of Drive for Thinness as one of the risk factors for the development of eating disorders in the group of Japanese women.

## 4. Discussion 

### 4.1. Similarities and Differences between Polish and Japanese Women

The results of this research suggest that Japanese women show a significantly higher level of bulimic tendencies (dependent variable) than Polish women. Other dependent variables—Body Dissatisfaction, Drive for Thinness, and restrictive behaviour towards nutrition and the body—did not differentiate the examined groups of Polish and Japanese women. This tendency may refer to “hattou shin ideal” phenomena among Japanese females. According to this social appearance construct, the ideal person is characterized by small head size and long legs [75]. As Nielson [76] indicates, the Asian women aspire to Western body standards, including being tall and thin. To strive for this ideal, Japanese women have engaged in obsessive exercise, vomiting or using laxatives [65,75,76]. Due to the lack of existing studies describing the aforementioned relations in populations of Japanese and Polish women, it is difficult to compare these results with other Polish and European studies. However, some papers provide results indicating the existence of a significant relation between dissatisfaction with body image and excessive pursuit of thinness and restrictive attitudes towards nutrition and the body (unhealthy and inappropriate for one’s BMI) evidenced by Japanese women [25,77]. It is worth mentioning that the similar restrictive tendencies and the presence of the correlation between them and body dissatisfaction are also observed in the population of young Polish women [5,78]. On the other hand, recent articles present less data on bulimic tendencies in the population of young Japanese women. Nevertheless, recent studies on bulimic behaviours were conducted on the population of Brazilian women and were published by de Carvalho et al. [28]. This topic has also been mentioned in research by Chapuis-de-Andrade et al. [40]. In the first of these studies, the authors examined a sample of 27,501 volunteers aged 18–55, whose average age was 28.9 ± 8.7 years. The research group consisted of 69.6% women. The results of this research indicate the relation between dysfunctional affective traits and the tendency for compensatory behaviours regarding food. A sample of 741 Brazilian women—female students with an average age 23.55 SD = 4.09—was examined by de Carvalho et al. [34]. Results of this research confirm the presence of a significant relation between increasing dissatisfaction with the body and compulsive eating behaviour. Furthermore, Agüera et al. [41] showed a similar relationship between emotional dysregulation, compulsive and bulimic eating behaviours, and nonacceptance of the body in their research conducted on Spanish women. The research review and meta-analysis of Prefit et al. [79] indicates a significant relation between emotional regulation and behaviours characteristic of eating disorders. Moreover, the second meta-analysis also showed the existence of a similar relation between emotional regulation and anorexia [80]. The lack of significant differences between Polish and Japanese women in the level of intensity of Body Dissatisfaction and Drive for Thinness demonstrated in this article can confirm the results of studies conducted by other authors, which indicate a universal and similar intensity of risk factors for the development of eating disorders in countries influenced by Western culture [1,3,8,11,81].

### 4.2. Predictors of Eating Disorders among the Studied Groups of Polish and Japanese Women

The results of the analysis of differences between Polish and Japanese women in terms of predictors of eating disorders, described in this article, indicate that among the psychological factors verified in the research model, Japanese women showed a significantly higher level of Emotional Dysregulation and Asceticism than Polish women. The results are interesting considering the Westernization effect. The Japanese media promote Western body ideal [75]. However, Japanese women were raised as delicate, restrained girls for years [65]. It may suggest that these attitudes are not so important among Polish women, despite the polish tradition based on Christian asceticism. Other research has shown the presence of similar risk factors of eating disorders in Asian countries, especially in Japan [25,72]. As previously mentioned, high levels of emotional dysregulation were related to eating disorders in studies conducted on a Spanish population [41] and on a Brazilian population of women [82]. Japanese women, to a greater extent than Polish women, search for information about body image in mass media. The level of intensification of internalization and pressures of socio-cultural standards of body image promoted by mass media proved to be similar in both examined groups. In both Polish and Japanese groups, the only equally important predictor of the Drive for Thinness variable was the global internalization of socio-cultural standards of body image promoted by mass media. The higher the intensity of Internalization-General, the stronger were thoughts and behaviours aimed at restrictively pursuing thinness [64,83,84].

Other significant predictors of Drive for Thinness were different for each group. Thus, it can be inferred that despite the cultural differences (“rice culture” vs. “potato culture”), there are some similarities which might be attributed to universal predictors between the studied women. Drive for Thinness is conditioned by the internalization of socio-cultural standards of body image and physical appearance promoted by mass media. Polish and Japanese women who pursue slimness seem to be under the influence of socio-cultural standards existing in mass media.

These data are consistent with the results of other studies conducted on various populations and nationalities, which indicate the increasing epidemiology of eating disorders around the world [1,19]. The results also confirm the significant role of socio-cultural predictors in the development of body dissatisfaction, which is one of the risk factors of eating disorders [2,39,70,78,85,86]. The results show that the significant predictors of variables such as Body Dissatisfaction and bulimic tendencies (Bulimia) for Japanese women were Emotional Dysregulation and Interoceptive Deficits—the former describes affective difficulties involving emotional dysregulation, such as emotional instability and impulsiveness, while the latter is related to disorientation and difficulties with identification and differentiation of emotional states and bodily sensations. These results may suggest that the intensity of Body Dissatisfaction in the group of Japanese women is explained mainly by emotional dysregulation and the directly-felt pressure of socio-cultural standards of body image promoted by mass media. On the other hand, bulimic tendencies in the group of Polish women are explained by Internalization-General and Personal Alienation, understood as existential difficulties related to feelings of emptiness and loneliness. The results show that the bulimic tendencies presented by Japanese women are explained by Emotional Dysregulation and Interpersonal Insecurity, which is related to difficulties in expressing and sharing personal thoughts and feelings with other people. In turn, Interpersonal Alienation, Interoceptive Deficits, and Internalization-General proved to be significant predictors of Body Dissatisfaction in the group of Polish women. They create a psychologically diverse system of emotional dysregulation, interpersonal difficulties, and socio-cultural internalization. Results of research conducted on the population of Brazilian and Spanish women emphasized the importance of emotional dysregulation in the development of bulimic and anorexic behaviours towards the nutrition and body [34,41]. The internalization of socio-cultural standards of body image existing in mass media is a universal predictor which explains all three examined factors related to risk of development of an eating disorder in the group of Polish women [5]. On the other hand, Internalization-Global was a predictor of Drive for Thinness and Pressures of socio-cultural standards explained the intensity of Body Dissatisfaction in the Japanese women. The results of this study show that the greater the internalization of socio-cultural norms of body image, the bigger are the interoceptive deficits and intensity of Asceticism presented by Polish and Japanese women. Polish women showed significantly higher positive correlation between these variables than Japanese women. The significance of this relation between these factors was higher in the group of Polish women. However, the significant correlation between the socio-cultural variable describing the frequency of seeking information about body image and Personal Alienation (sense of emptiness and loneliness) and Asceticism was stronger in the group of Polish women than in the group of Japanese women. It is also worth mentioning that the aforementioned relation between variables has a different direction for Japanese women. The results of this study show that the higher the intensity of Personal Alienation and sense of emptiness and the greater the Asceticism demonstrated by Japanese women, the lower their frequency of seeking information about body image. Asceticism and Personal Alienation reduce the frequency of searching for information about physical appearance in the population of Japanese women. However, the same variables increase the frequency of seeking information about body image in the group of Polish women. It seems that feelings of personal alienation and high asceticism diminish the need for searching for information about body image in Japanese women. In the case of Polish women, however, both personal alienation and feelings of emptiness and asceticism increase the frequency of reaching for information regarding physical appearance. Thus, it appears that emotional attitude and alienation can influence seeking information in mass media about body image in different directions. These innovative and interesting results suggest the existence of two culturally different ways of reacting. Polish women have a tendency to react more impulsively. On the other hand, Japanese women tend to react by withdrawing from seeking further information about body image. 

An explanation of the aforementioned correlations can be found in the upbringing environment and the culture which influenced it. Internalization of observed values and principles is the consequence of the influence of one’s surrounding culture and upbringing environment. Japanese women grew up in an entirely different culture than did Polish women. Japanese women tend to be more subordinated to principles of traditional social and family life. On the other hand, the desire for European beauty is particularly strong among Asian women [87]. However, Japanese women may tend to be less individualistic than the European women [88]. Comparing this conclusion with the obtained results, it can be assumed that the more Japanese women are characterized by personal, traditional alienation and asceticism, the more consciously they avoid seeking information about body image in mass media. Polish women grew up in different cultural conditions, with less emphasis on compliance with traditional norms than Japanese women. On the basis of the obtained results, it can be suggested that the greater the sense of emptiness and alienation and asceticism, the more consciously Polish women will search for information about physical appearance in mass media. The different traditions of childrearing and the tendency to subordinate characteristic to each culture also manifested in other relationships between the studied variables. The next issue worth mentioning is the relationship between the unconsciously internalized standards of body image and the Drive for Thinness. These relations are similar between both Polish and Japanese women. The results show the presence of the same correlation in both groups: The higher the Drive for Thinness, the greater the global internalization. This confirms the impact of Western culture which promotes the “cult of thinness” in both populations of young Polish [78] and Japanese women [25,77].

Analysis of psychological predictors of bulimic behaviours in Polish and Japanese women showed some differences between the groups. Bulimic tendencies in the group of Japanese women are explained by many emotional variables such as Emotional Dysregulation, Interoceptive Deficits, Asceticism, and Interpersonal Insecurity. However, the strongest predictors of the development of bulimic behaviours in the group of Polish women were Personal Alienation, BMI, and Internalization-General of standards of body image existing in mass media. Thus, predictors of bulimic behaviours in the group of Polish women are variables related primarily to the level of BMI, Personal Alienation—a sense of emotional emptiness which can be related to depression (the strongest predictor)—and the impact of the internalization of socio-cultural standards of body image present in mass media. Other psychological factors proved to be irrelevant as predictors for bulimia. When explaining bulimic tendencies presented by Japanese women, it is worth considering a set of psychological variables describing emotional attitude towards oneself and interpersonal relationships. In the population of Polish women, the main predictors for bulimic behaviours were: Personal Alienation, BMI, and unconscious internalization of socio-cultural standards. Results of analysis of stepwise regression show differences in the distribution, significance, and type of predictors of Body Dissatisfaction between the groups of Japanese and Polish women. Interoceptive Deficits proved to be a significant predictor of Body Dissatisfaction in both groups. Nevertheless, the strength of the relationship between the variables was positive in the group of Polish women and negative in the group of Japanese women. 

A probable explanation of the differences between these two groups is the presence of another significant predictor of Body Dissatisfaction in the group of Japanese women: Emotional Dysregulation. This variable proved to be irrelevant in the prediction of dependent variables in the group of Polish women. The third significant predictor of Emotional Dysregulation in the group of Japanese women is Pressures of socio-cultural standards. Perhaps the co-occurrence of two predictors concerning emotional state (i.e., Interoceptive Deficits and Emotional Dysregulation) and the socio-cultural predictor (Pressures of socio-cultural standards) explain the differences between the groups of Japanese and Polish women in terms of the significance of the relation between the variables. 

The analysis of the obtained results suggests that educational and preventive interventions (in the context of bulimic and restrictive eating behaviours) should focus on developing the effective body standards and understand the ideal appearance standards, promoted in West Cultures. The importance of such interventions were indicated by Nelson et al. [75]. The authors suggest incorporating such interventions into health school programs. 

The main practical implications of the authors’ research relate to: (a)the need of psychological diagnosis: sociocultural and psychological predictors (risk factors) of eating disorders.

The measurement of these factors may be necessary in the practice of family physicians, psychiatrists, pediatricians, educators and other specialists who support adolescents’ optimal health and development (especially in developing positive body image). According do prevention, discussing the Western body ideals and the role of Westernization in developing eating disorders should be obvious. 

(b)taking into account the predicators of eating disorders in prevention programmes towards anti-health eating behaviours (restrictive and bulimic) and body image in adolescents and young women (c)taking into account the predictors of anti-health eating behaviours among adolescents and young people and body image in the therapeutic programmes conducted by psychologists and psychotherapists(d)taking into account the results of the authors’ research in the process of creating local crisis support in which socio-cultural and psychological factors of developing eating disorders will be quickly detected.

### 4.3. Limitations of the Study 

The presented research was a rare or even unique opportunity to make cultural comparisons between two different nationalities. However, the results of this study are also subject to some limitations. The research groups, especially the Japanese group, although sufficient in number for statistical analyses, were not very large. However, the analyses gave the opportunity to make comparisons which were previously lacking in the literature between two clearly different cultures and attitudes towards nutrition (“rice culture” vs. “potato culture”). It should be also noted that Japan is very geographically distant from Poland, so access to the local population was limited. Statistical analysis showed that the distribution of the examined variables did not meet the standards of a normal distribution. Although the aim of this study was to identify specific cultural differences, it should be noted that both groups were of a similar, narrow age-range. Perhaps an extension of this research to older populations would lead to different results in the intensity of existing differences and relations between the examined groups. 

## 5. Conclusions

Intercultural differences between the studied groups of Japanese and Polish women show that Japanese women present higher intensity of the dependent variable Bulimia and independent variables such as Emotional Dysregulation, Asceticism, Maturity Fears, and frequency of seeking information about body image. 

Results of the study indicate the presence of a similarity between Polish and Japanese women in terms of high levels of internalization and pressures due to socio-cultural body image standards promoted by mass media. There were no significant intercultural differences in intensity of those variables. 

The global internalization of socio-cultural standards of body image proved to be a significant predictor of the Body Dissatisfaction variable in the groups of Polish and Japanese women. Due to this result, it can be confirmed that Internalization-General is a universal socio-cultural predictor of dissatisfaction with one’s body in both groups of women. 

Emotional Dysregulation, Interoceptive Deficits, Asceticism, and Pressures of socio-cultural standards are statistically significant predictors of Drive for Thinness and Bulimia in the group of Japanese women. Some of these predictors proved to be irrelevant in explaining the dependent variables for the group of Polish women. 

The results of the analyses show that the main predictors obtained for Polish women describe a set of interpersonal and existential difficulties associated with feelings of emptiness and loneliness. Moreover, they indicate the significant role of emotional predictors associated with excessive Asceticism, Emotional Dysregulation, and Interoceptive Deficits. 

The dominant psychological predictors of Body Dissatisfaction and Bulimic behaviour in the group of Japanese women are Emotional Dysregulation, Asceticism, and Interoceptive Deficits. In turn, Personal and Interpersonal Alienation were significant predictors of Body Dissatisfaction and Bulimic tendencies in the group of Polish women. 

However, it is difficult to compare the obtained results with other studies due to the lack of comparative research on populations of Polish and Japanese women of the same age. 

## Figures and Tables

**Table 1 nutrients-12-00666-t001:** Comparative analysis between groups of Japanese (*n* = 70) and Polish (*n* = 89) women in terms of the variables included in the research model.

Feature		Polish Women				Japanese Women				Differences		
		*M*	min	max	*SD*	*M*	min	max	*SD*	*U*	*Z*	*p*
EDI-3	Bulimia	5.47	0.00	32.00	6.07	8.8	0.00	28.00	6.59	2084.00	−3.58	0.001
	Body Dissatisfaction	17	0.00	40.00	10.88	16.6	0.00	38.00	7.92	3020.00	0.33	0.743
	Drive for Thinness	9.88	0.00	28.00	8.48	10.62	0.00	22.00	5.89	2708.00	−1.41	0.158
	Low Self-Esteem	8.92	0.00	23.00	6.67	8.26	1.00	15.00	3.43	2979.00	0.47	0.638
	Personal Alienation	9.34	0.00	28.00	7.10	8.54	0.00	17.00	3.78	3068.00	0.16	0.872
	Interpersonal Insecurity	10.57	0.00	26.00	6.56	9.26	0.00	21.00	4.78	2792.50	1.12	0.264
	Interpersonal Alienation	8.78	0.00	27.00	6.05	9.51	0.00	20.00	3.92	2730.00	−1.33	0.182
	Interoceptive Deficits	10.76	0.00	35.00	8.69	10.9	0.00	25.00	6.52	2894.00	−0.77	0.444
	Emotional Dysregulation	6.75	0.00	32.00	5.77	8.93	0.00	23.00	5.77	2365.00	−2.60	0.009
	Perfectionism	10.19	0.00	24.00	5.57	9.3	1.00	21.00	4.03	2917.50	0.68	0.494
	Asceticism	7.15	0.00	23.00	5.32	8.17	1.00	22.00	4.79	2593.00	−1.81	0.035
	Maturity Fears	11.56	0.00	31.00	7.53	12.24	4.00	23.00	4.54	2622.50	−1.71	0.044
SATAQ 3	Information	22.3	9.00	43.00	7.82	28.13	9.00	43.00	6.12	1631.50	−5.15	0.001
	Pressures	18.55	7.00	32.00	7.27	20.54	7.00	31.00	5.81	2666.00	−1.56	0.120
	Internalization-Athlete	14.02	5.00	25.00	5.19	13.97	5.00	23.00	3.48	3030.50	−0.29	0.771
	Internalization-General	25.52	9.00	42.00	8.98	27.24	11.00	42.00	6.30	2880.50	−0.81	0.417

Note: EDI-3—The Eating Disorders Inventory, SATAQ 3—The Sociocultural Attitudes Towards Appearance Questionnaire.

**Table 2 nutrients-12-00666-t002:** Correlation analysis for all variables for the groups of Polish and Japanese women (Spearman’s rho coefficient).

EDI-3		SATAQ3			
		Information	Pressures	Internalization Athlete	Internalization General
Bulimia	PL	0.02	0.14	0.17	0.25 *
	JAP	−0.02	0.34 **	0.28 *	0.18
Body Dissatisfaction	PL	0.07	0.33 **	0.34 ***	0.41 ***
	JAP	0.25 *	0.51 ***	0.24 *	0.44 ***
Low Self-Esteem	PL	0.12	0.20	0.22 *	0.31 **
	JAP	0.06	0.09	0.23 *	0.15
Personal Alienation	PL	0.15	0.17	0.22 *	0.31 **
	JAP	−0.23 *	0.03	0.15	−0.09
Interpersonal Insecurity	PL	0.12	0.13	0.14	0.20
	JAP	−0.09	0.01	0.19	0.01
Interpersonal Alienation	PL	0.15	0.15	0.19	0.25 *
	JAP	−0.04	0.23 *	0.16	0.20
Interoceptive Deficits	PL	0.16	0.18	0.25 *	0.36 ***
	JAP	−0.20	0.24 *	0.15	0.09
Emotional Dysregulation	PL	0.16	0.17	0.18	0.39 ***
	JAP	−0.05	0.18	0.13	0.20
Perfectionism	PL	0.08	0.26 *	0.31 **	0.38 ***
	JAP	0.13	0.21	0.24 *	0.23 *
Drive for Thinness	PL	0.24 *	0.37 ***	0.47 ***	0.54 ***
	JAP	0.12	0.56 ***	0.35 **	0.43 ***
Asceticism	PL	0.25 *	0.31 **	0.38 ***	0.44 ***
	JAP	−0.10	0.34 **	0.25 *	0.14
Maturity Fears	PL	−0.02	0.12	0.10	0.24 *
	JAP	−0.10	0.14	0.13	0.22

Note: EDI-3—The Eating Disorders Inventory; SATAQ 3—The Sociocultural Attitudes Towards Appearance Questionnaire; * *p* < 0.05; ** *p* < 0.01; *** *p* < 0.001; PL: Polish Women; JAP: Japanese Women.

**Table 3 nutrients-12-00666-t003:** Comparison of predictors of risk factors for development of eating disorders in the groups of Polish and Japanese women.

Dependent Variable	Polish Women	Japanese Women
Bulimia	Adjusted R^2^ = 0.312 F(4.84) = 11.01 *** Predictors: Personal Alienation B = 0.562 *** BMI B = 0.231 * Internalization General B = 0.189 *	Adjusted R^2^ = 0.609 F(7.62) = 16.38 *** Predictors: Asceticism B = 0.270 * Interoceptive Deficits B = 0.306 * Emotional Dysregulation B = 0.236 * BMI B = 0.163 * Interpersonal Insecurity B = -0.203 *
Body Dissatisfaction	Adjusted R^2^= 0.589 F (6.82) = 22.02 *** Predictors: Interoceptive Deficits B = 0.291 ** BMI B = 0.295 *** Internalization General B = 0.248 * Interpersonal Alienation B = 0.340 *** Information B = −0.177 *	Adjusted R^2^ = 0.452 F (6.63) = 10.48 *** Predictors: Pressures B = 0.476 *** Emotional Dysregulation B = 0.398 ** BMI B = 0.211 * Interoceptive Deficit B = −0.372 **
Drive for Thinness	Adjusted R^2^ = 0.611 F (3.85) = 47.09 *** Predictors: Asceticism B = 0.556 *** Internalization General B = 0.301 ***	Adjusted R^2^ = 0.543 F (6.63) = 14.07 *** Predictors: Interoceptive Deficits B = 0.380 *** BMI B = 0.260 * Internalization General B = -0.253 **

Note: * *p* < 0.05; ** *p* < 0.01; *** *p* < 0.001.

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
