# Peer review of "Body Dissatisfaction, Restrictive, and Bulimic Behaviours among Young Women: A Polish–Japanese Comparison"

_nutrients, 2020, doi:10.3390/nu12030666_

Round 1
Reviewer 1 Report
The submitted manuscript examine the psychological and socio-cultural risk factors for eating disorders in Polish and Japanese women.
In Discussion section: Authors must avoid the repetition of the research questions that are included in the first section: introduction.
Tables 1 and 2:The acronyms (EDIT-3; SATAQ3; PL, JAP) that appear in the tables should be explained at the bottom of the table.
References: Adapt references to the rules about the number of authors cited: Author 1, A.B.; Author 2, C.D. Title of the article. Abbreviated Journal Name Year, Volume, page range.
Author Response
We would like to thank the Reviewer for all the remarks and comments on our manuscript. We proceeded with the revision of the manuscript taking into account all the comments and the issues raised. Below we explain in detail how we addressed those issues.
Reviewer’s comment:
In Discussion section: Authors must avoid the repetition of the research questions that are included in the first section: introduction.
Response:
As suggested by the reviewer, the research questions have been removed from the Discussion section.
Reviewer’s comment:
Tables 1 and 2:The acronyms (EDIT-3; SATAQ3; PL, JAP) that appear in the tables should be explained at the bottom of the table.
Response:
We have added this information.
Reviewer’s comment:
References: Adapt references to the rules about the number of authors cited: Author 1, A.B.; Author 2, C.D. Title of the article. Abbreviated Journal Name Year, Volume, page range.
Response:
We have carefully checked the bibliographic records and adapted it to the requirements of the Journal (ASC).
Reviewer 2 Report
Comments to the Author
The research has addressed a very interesting research question and puts the cultural element into perspective.I think the manuscript could be greatly improved if the authors took time to justify how the two cultures are different and what is the impact of the context on body image. Finally, I would like to see further information in the introduction on the internalization of socio-cultural standards of body image universally. This would help to tie in information about Polish and Japanese women. The discussion could focus more on the applied implications of the results.When reading the study the originality and significance of the research does not come out as clearly as it could. In this regard I think the authors would do well to be more critical of previous work, as well as making the novelty of their findings much clearer. The manuscript could be greatly improved if the authors took time to explain what the unique socio-cultural pressures in Japan and Poland are in the discussion and how these relate to the rest of the world based on research that has already been done.
Please see below for some specific comments:
- Line 61- In the section about risk factors for eating disorders, I think the socio-cultural risk factors have not really been emphasized as much as I would expect given the nature of this study.
- Line 65- Please add reference
- Line 81-83 – Please add reference
- It is not clear if there is existing research on the Japanese and Polish culture on eating disorders. This should be put straight. I would like to have seen
- Line 214- How many cities in Japan? Is it not important to know the demographics. Perhaps, you can move the names of the cities from line 230. Even if you do not state which cities at this point you should at least say how many cities?
- Line 216- It is still unclear how the participants were recruited? How did you find them? Did you use flyers? Where did you recruit them from?
- Line 221- How did you decide this was an adequate sample size? Did you calculate the effect size? Perhaps you could mention your rationale and justify your decision. Sample sizecalculation is an essential item to be included in the paper.
- Have a look at some more literature that ate specific for Japan and Poland such as Chisuwa, N., & O’Dea, J. A. (2010). Body image and eating disorders amongst Japanese adolescents. A review of the literature. Appetite, 54(1), 5-15. Wlodarczyk‐Bisaga, K., Dolan, B., McCluskey, S., & Lacey, H. (1995). Disordered eating behaviour and attitudes towards weight and shape in Polish women. European Eating Disorders Review, 3(4), 205-216.
- How does your study compare with other cross-cultural studies? Are there differences in the societal beauty ideals? For example, look at the study by Swami, V., Caprario, C., Tovée, M. J., & Furnham, A. (2006). Female physical attractiveness in Britain and Japan: a cross‐cultural study. European Journal of Personality: Published for the European Association of Personality Psychology, 20(1), 69-81.
Author Response
We would like to thank the Reviewer for all the remarks and comments on our manuscript. We proceeded with the revision of the manuscript taking into account all the comments and the issues raised. In attachment we explain in detail how we addressed those issues.

Round 2
Reviewer 2 Report
I appreciate that you made an effort to address all the comments carefully. I think that you did a good job, however some further changes can significantly improve the manuscript. For example. the implications for practice have only been briefly discussed. I would expect them to be spread throughout the discussion section instead of a small paragraph added at the end.
Another concern I have is that the rationale for the study did not fully and critically engage with contemporary literature on body image in different cultures.
Finally there is a concern for methodological rigour, particularly as this related to the analysis and presentation of data, which were not interpreted/tied back to a more extensive discussion. In turn, the discussion did not fully develop and clarify the novelty and contribution of the research as to how the literature on westernalization and body image was expanded beyond the "same old".
Author Response
We would like to thank the Reviewer for the further comments on our manuscript. We proceeded with the revision of the manuscript taking into account all the comments. Below we explain in detail how we addressed those issues.
Reviewer’s comment:
I appreciate that you made an effort to address all the comments carefully. I think that you did a good job, however some further changes can significantly improve the manuscript.
For example. the implications for practice have only been briefly discussed. I would expect them to be spread throughout the discussion section instead of a small paragraph added at the end.
Response:
As suggested by the reviewer, the implications for practice have been added (tracked changes).
Reviewer’s comment:
Another concern I have is that the rationale for the study did not fully and critically engage with contemporary literature on body image in different cultures.
Response:
We have added this newest literature (tracked changes – lines 67-74 and Reference- five new bibliographic records).
Reviewer’s comment:
Finally there is a concern for methodological rigour, particularly as this related to the analysis and presentation of data, which were not interpreted/tied back to a more extensive discussion. In turn, the discussion did not fully develop and clarify the novelty and contribution of the research as to how the literature on westernalization and body image was expanded beyond the "same old".
Response:
We have added new information, ideas, and suggestions in Discussion (lines 488-493, 524-527, 588-589, 596-599; three new bibliographic records).